# Impact of pH and High-Pressure Pasteurization on the Germination and Development of *Clostridium perfringens* Spores under Hyperbaric Storage versus Refrigeration

**DOI:** 10.3390/foods13121832

**Published:** 2024-06-11

**Authors:** Carlos A. Pinto, Alireza Mousakhani Ganjeh, Francisco J. Barba, Jorge A. Saraiva

**Affiliations:** 1Associated Laboratory for Green Chemistry of the Network of Chemistry and Technology (LAQV-REQUIMTE), Department of Chemistry, University of Aveiro, 3810-193 Aveiro, Portugal; carlospinto@ua.pt (C.A.P.); mousakhani@ua.pt (A.M.G.); 2Research Group in Innovative Technologies for Sustainable Food (ALISOST), Preventive Medicine and Public Health, Food Science, Toxicology and Forensic Medicine Department, Faculty of Pharmacy, Universitat de València, Avda. Vicent Andrés Estellés, s/n, 46100 Burjassot, Spain; francisco.barba@uv.es

**Keywords:** *Clostridium perfringens* spores, hyperbaric storage, refrigeration, coconut water, BHI-broth, food safety

## Abstract

This study aimed to evaluate hyperbaric storage at room temperature (75–200 MPa, 30 days, 18–23 °C, HS/RT) on *Clostridium perfringens* spores in brain-heart infusion broth (BHI-broth) at pH 4.50, 6.00, and 7.50 and coconut water (pH 5.40). Both matrices were also pasteurized by high pressure processing (600 MPa, 3 min, 17 °C, HPP) to simulate commercial pasteurization followed by HS, in comparison with refrigeration (5 °C, RF). The results showed that, at AP/RT, spores’ development occurred, except at pH 4.50 in BHI-broth, while for RF, no changes occurred along storage. Under HS, at pH 4.50, neither spore development nor inactivation occurred, while at pH 6.00/7.50, inactivation occurred (≈2.0 and 1.0 logs at 200 MPa, respectively). Coconut water at AP/RT faced an increase of 1.6 logs of *C. perfringens* spores after 15 days, while for RF, no spore development occurred, while the inactivation of spores under HS happened (≈3 logs at 200 MPa). HPP prior to HS seems to promote a subsequent inactivation of *C. perfringens* spores in BHI-broth at pH 4.50, which is less evident for other pHs. For HPP coconut water, the inactivation levels under HS were lower (≈2.0 logs at 200 MPa). The Weibull model well described the inactivation pattern observed. These results suggest that HS/RT can be simultaneously used as a tool to avoid *C. perfringens* spores’ development, as well as for its inactivation, without the application of high temperatures that are required to inactivate these spores.

## 1. Introduction

*Clostridium perfringens* is a Gram-positive, anaerobic bacterium that forms spores, which are able to germinate at temperatures between 12 and 46 °C and pHs between 5 and 8 [1]. This pathogenic microorganism is responsible for causing gastrointestinal ailments in both humans and animals, with the specific variant recognized for inducing *C. perfringens*-linked food poisoning in humans being *C. perfringens* type F [2]. For example, between 1998 and 2010, the United States Center for Disease Control and Prevention confirmed 289 outbreaks, resulting in over 15,000 illnesses in the United States [3], while in Singapore, both *Salmonella* spp. and *C. perfringens* instigated 171 outbreaks involving 7538 cases of illness [4].

The spores of *C. perfringens* are quite resistant against inactivation such as by high temperatures (below 100 °C), hydrostatic pressures, desiccation, ultraviolet radiation, etc., allowing them to survive, although in a dormant form, for extended periods of time [5]. This thermal resistance can vary considerably depending on the ability of the strains to produce enterotoxins, since *C. perfringens* spore strains encoding for enterotoxin production (cpe gene) exhibit higher heat resistance, presenting D-values of 53.2 min, while spores from non-CPE-producing strains are more heat-sensitive, presenting a D-value of about 1.0 min (both at 99 °C).

Upon encountering favorable conditions, these spores can end their dormant state and trigger the germination process. This phenomenon is prompted by a range of compounds (nutrients) referred to as germinants, such as amino acids, a combination of calcium-chelates and pyridine-2,6-dicarboxylic acid (known as dipicolinic acid, DPA), alongside other non-nutrient substances like dodecylamine, which functions as a cationic surfactant, or even by physical stimulus, such as low/moderate hydrostatic pressures [6,7]. The spores of *C. perfringens* can occasionally be found in processed meat products (hamburgers, sausages, minced meat), ready-to-eat meals, sausages, or even on the surface of vegetables [8], resisting commercial pasteurization processes, like thermal and HPP. The most used strategies used for hurdling *C. perfringens* spores in raw and pasteurized foods include the use of cold storage (RF or freezing), osmotic-induced stress (dehydration), chemical preservatives such as nitrites, organic acids (sorbic, benzoic, lactic, acetic acids, for pH depletion ≤ 4.6), etc. [9]. Pasteurization targets vegetative pathogenic microorganisms, with spores not being inactivated, and this is particularly important for low-acidic foods (pH > 4.6). So, in order to inhibit spore germination, the food products pasteurized by HPP are to be kept under RF [10].

Coconut water, a milky liquid derived from coconut fruit, is considered a nutritious beverage due to its high content of calcium, magnesium, vitamin B, and vitamin C. It is often consumed in tropical regions for its distinct taste that is appreciated by consumers [11]. According to its maturity state, coconut water can be considered as “tender” when derived from green coconuts or “mature” coconut water when derived from older coconuts, which affects the pH [12]. In fact, the pH varies according to the maturity state, being around 4.9–5.2 for tender coconut water and >5.2–6.3 for mature coconut water [13,14,15]. As it is a quite perishable food product, raw coconut water has a very short shelf-life, and thermally pasteurized changes considerably affect the organoleptic properties [16]. In 2015, the Food and Drug Administration (FDA) questioned the use of HPP for nonthermal pasteurization of low-acidic coconut water, addressing safety concerns towards the possibility of *Clostridium botulinum* spores’ germination and development under RF conditions, as HPP cannot destroy spores [17]. As such, companies ended up stopping the commercialization of low-acidic coconut water due to the food safety concerns raised by the FDA. This topic has raised interest, especially from the food industry, who rushed to try to prove that *C. botulinum* spores do not develop in coconut water [18]. However, the situation is not yet clear whether *C. botulinum* spores can or not develop in coconut water.

Recently, a new food preservation methodology termed HS was proposed as a method not only to inhibit endospore germination and outgrowth but also to inactivate bacterial spores. This methodology uses storage pressure control (between 75 and 220 MPa, similar to conventional RF that states temperature control) to hurdle microbial development. HS at RT can attain considerable energetic savings, as energy is only mobilized during the short compression and decompression phases of the pressure vessel, and no additional energy is needed to keep the foods stored under pressure. As a matter of fact, RF is responsible for about 17% of the total electricity consumed worldwide, and about 8% of the greenhouse gas emissions [19].

Previous studies have shown that HS/RT is able not only to avoid spores development in commercial apple juice but also to cause inactivation (4.5 log units after 30 days at 100 MPa for *Bacillus subtilis* and 5 log units after 48 h at 100 MPa for *Alicyclobacillus acidoterrestris*), even though the products were kept at uncontrolled RT [20,21]. Other studies in fresh Atlantic salmon and raw milk have also suggested that HS/RT seems to be a feasible food storage methodology to avoid *B. subtilis* spores’ development.

Nevertheless, the feasibility of HS at uncontrolled RT in *Clostridium* spp. spores is not reported in the literature and, considering the safety issues related to the presence of *C. perfringens* spores in foods and their ability to survive pasteurization methodologies, such as thermal and HPP pasteurization, this study aimed to evaluate the performance of HS (75, 150, and 200 MPa) at uncontrolled RT (18–23 °C) to control the development of *C. perfringens* spores, using BHI-broth as a model system, at three different pH levels (4.50, 6.00 and 7.50). The results obtained were then used for a more finetuned study in a real food product, using coconut water (pH 5.40) as a case study. Additionally, in order to simulate commercial pasteurization conditions, the BHI-broth model and coconut water were inoculated with *C. perfringens* spores and pasteurized by HPP (600 MPa, 3 min, 17 °C) and then stored at the same HS conditions aforementioned. The selection of coconut water as a validation case study relies on the previously mentioned facts that, due to its highly perishability, it does not possess natural hurdles against spores’ development; as such, it has a very short shelf-life when it is unprocessed (raw), and RF can only temporarily delay the development of spores (as it was issued by the FDA in 2015). As such, this is an iconic case study and a good candidate to study the effects of a new preservation methodology (HS/RT) on *C. perfringens* spores.

## 2. Materials and Methods

### 2.1. Culture Media and Chemicals

BHI-broth, BHI-agar, cooked meat medium, *Clostridium perfringens* sporulation broth and thioglycolate broth (Himedia, Thane, India), Anaerocult^TM^ A (Merck, Darmstadt, Germany), citric acid, physiological solution (consisting of 0.9% NaCl), and Applichem Panreac (Darmstadt, Germany) wrere used in this study.

### 2.2. Spores’ Production, Harvesting, Storage, and Inoculation

The *C. perfringens* spores were prepared as described in [22] with slight modifications. A single colony of a pure culture of *C. perfringens* NCTC 8237 (ATCC 13124) was inoculated in test tubes containing sterilized cooked meat medium and incubated under anaerobic conditions using an anaerobic jar with a Anaerocult^TM^ A patch at 37 °C for 48 h. Then, the test tubes were homogenized, and an aliquot of 0.1 mL was transferred to 9.9 mL of thioglycolate broth and incubated under anaerobic conditions at 37 °C for 8 h. Afterwards, 0.2 mL of the culture was transferred to test tubes containing 9.8 mL of *C. perfringens* sporulation broth and then incubated at 28 °C for 12 days. This sub-optimal temperature was selected to increase the sporulation rates of *C. perfringens* [23]. Then, the spores were washed three times by centrifugation (4000× *g*, 10 min, 4 °C) in cold, sterilized, distilled water. The spores used in the present study were not heat-treated prior to inoculation to avoid changes on the pressure resistance of the spores [24].

The spores were then inoculated in heat-sterilized (121.1 °C for 15 min in an autoclave) BHI-broth, adjusted to pH 4.50, 6.00, and 7.50, and filter-sterilized (0.22 µm) coconut water (to avoid chemical changes induced by the intense thermal processing), with pH 5.40. Both matrices were packed under aseptic conditions in microcentrifuge polypropylene tubes with 0.4 mL capacity (Beckman Coulter, Brea, CA, USA), which were previously sterilized by UV-radiation for 15 min, using a laminar flow chamber (BioSafety Cabinet Telstar Bio II Advance, Terrassa, Spain) to avoid contaminations. A final concentration of spores ranging between 10^5^ and 10^6^ spores/mL was obtained.

### 2.3. HPP Pasteurization

In order to study the effects of a previous commercial-like nonthermal pasteurization by HPP on the subsequent evolution of *C. perfringens* spores under HS conditions, both the inoculated BHI-broth (at pH 4.50, 6.00, and 7.50) and coconut water (pH 5.40) were subjected to nonthermal HPP pasteurization (600 MPa, 3 min, 17 °C), using pilot-scale high-pressure equipment (Hiperbaric 55, Hiperbaric S.A., Burgos, Spain) to simulate commercial-like processing conditions by HPP.

### 2.4. Storage Conditions

The HS conditions were set at 75, 150, and 200 MPa at uncontrolled room temperature (18–23 °C), with the samples being placed in a high-pressure equipment (FPG13900, Stansted Fluid Power, Stansted, UK) for up to 30 days. This equipment has a pressure vessel of 35 mm inner diameter and 400 mm height and uses a mixture of water and propylene glycol (40:60 *v*/*v*) as pressurization fluid. Simultaneously, control samples were kept at atmospheric pressure (0.1 MPa, AP) at RT and under RF (5 °C), which were immersed in the same pressurization fluid and kept in the dark.

### 2.5. Determination of Spores’ Germination and Inactivation

To evaluate the spores after each storage condition, germinated (partially germinated spores and vegetative microorganisms, and non-germinated (termed total microbial load, TML) and only non-germinated (called heat resistant endospore fraction, HREF) spores were assessed. In this sense, right after each storage period, 0.1 mL of each sample was serially diluted in 0.9 mL of physiological solution (0.90% NaCl) and spread-plated in BHI-agar plates using the Miles and Misra technique [25]. The plates were then incubated at 37 °C for 24 h under anaerobic conditions (in anaerobic jars). Petri dishes containing between 1 and 100 colony-forming units (CFU) were selected for counting (this procedure allowed us to determine TML). Then, an aliquot of each matrix was heat-treated at 70 °C for 10 min to inactivate the vegetative forms, leaving only the non-germinated spores (HREF) [20,26,27] and then plated and incubated as aforementioned. The results were expressed as decimal logarithm variation (Log N/N_0_), given by the difference between the microbial loads at a specific storage day (N) and the initial microbial load (N_0_).

### 2.6. Spore Inactivation Kinetics Modelling

Two different mathematical models were used to describe spores’ inactivation kinetics, the first-order and the Weibull models, respectively, displayed in Equations (1) and (2). The data fitting was performed using the software Matlab R2022a (MathWorks Inc., Natick, MA, USA) to fit the spore survival curves and estimate the parameters of each model (only quantifiable experimental values were used for the model fitting, i.e., values below quantification and detection limits values were not used). The mean square root error (*MSRE*), coefficient of determination (*R*^2^), and adjusted-*R*^2^ (adj-*R*^2^) were determined to infer the fitting quality of the model. A *MSRE* close to 0 and a *R*^2^ and adj-*R*^2^ close to 1 indicates the adequacy of the models to describe the experimental data. The values estimated for each model parameter are displayed as value ± 95% confidence interval.

#### 2.6.1. First-Order Kinetic Model

The decimal reduction time (*D_t_*-values, time required at a certain pressure to reduce the microbial population by 90%), was determined from the reciprocal of the slope [28], according to Equation (1):(1)Log NN0=⁡−tDt
where *N*_0_ and *N* regard the initial spore population in each matrix and the number of survivors after being exposed under lethal HS conditions for a certain period of time.

#### 2.6.2. Weibull Model

The Weibull model is used to describe inactivation kinetics of spores, as reported in [29], and Equation (2) was used to fit the experimental data (microbial load reduction (log *N*/*N*_0_) along storage:(2)Log NN0=⁡−btn
where *b* regards the scale factor parameter and is related to the velocity of the microbial inactivation along storage (giving information about the dying behavior of the spores), n is the survival curve shape factor [30], determining this parameter the shape of the curve, being concave-upwards (tailings) if *n* < 1 and concave-downwards (shoulders) if *n* > 1 (*n* = 1 indicates a simpler first-order kinetics).

### 2.7. Statistical Analysis

The performed analyses were performed in duplicate, each one from triplicated samples. The obtained results are presented as mean ± standard deviation and were statistically analyzed by one-way Analysis of Variance (ANOVA), followed by Tukey’s Honestly Significant Difference (HSD) test at a significance of 5%.

## 3. Results and Discussion

### 3.1. Impact of pH on the C. perfringens Spores’ Behavior under HS

The initial TML and HREF (both at around 5–6 logs, except for HPP pasteurized coconut water for which the values were 4.79 and 3.33, respectively) are available in Appendix A. At pH 4.50 (Figure 1), neither endospore germination nor outgrowth was observed at atmospheric pressure and room temperature (AP/RT) and under RF, as expected, as this pH was below the 4.6 threshold required to inhibit *C. botulinum* spores germination [31]. Nevertheless, a tendency for a slight decrease was observed for HS, more noticeable for 150 and 200 MPa, with a small (*p* < 0.05) TML decrease being observed (0.47 and 0.70 log units, respectively), and a similar trend was observed for HREF. This can be attributed to the fact that at such pH level, most of the spores’ proteins are protonated (considering that the isoelectric point of most spores’ proteins range between 5–6.5), which do not allow for a proper trigger of the germination and outgrowth process [32]. As such, these pH hinders the germination triggered by low hydrostatic pressures (nutrient-like physiological germination) [26].

It is also noteworthy that the thermal resistance of *C. perfringens* spores is quite dependent on their ability to produce enterotoxins, i.e., *C. perfringens* enterotoxigenic (cpe) strains (those carrying the chromosomal cpe gene) are known to have higher thermal resistance than those lacking the cpe gene. This resistance is primarily due to the presence of a variant of the small, acid-soluble protein (Ssp4), which binds strongly to spore DNA, protecting it from heat damage [33].

Differently, at pH 6.00 and AP/RT conditions, we observed a clear trend with an increase (*p* < 0.05) of both TML and HREF of 1–2 logs, while no noticeable changes were observed for TML and HREF under RF (Figure 2). In a different way, for HS, a clear decay pattern was observed for both the TML and HREF, increasing gradually with the HS pressure increment, with inactivation reaching after 30 days at 75, 150, and 200 MPa, with values being reduced by 2.01, 1.25, and 1.75 log units for TML, respectively, while the HREF was reduced by 1.66, 1.07, and 1.75 log units, respectively. The inactivation verified by HS may be related to the occurrence of the nutrient-like physiological germination, a germination mechanism that occurs under low/moderate hydrostatic and reported to optimal for pressures around 150–200 MPa for *B. subtilis* spores, while *Clostridium* spp. spores seem to be less affected at such pressure levels, as observed by [34], who reported very mild changes on the germination levels (measured by DPA release) of *C. perfringens* MRS101 spores after 15 min under 150 MPa at 37 °C, contrarily to *B. subtilis* spores, whose germination rates were higher than 95% after 8 min under 150 MPa at 37 °C in 50 mM ACES buffer at pH 7.00 [35], suggesting that *C. perfringens* spores may respond differently to pressure compared to *B. subtilis* spores. In a study by [36] with *Clostridium sporogenes* inoculated in minced anchovies and kept under pressure at 50 MPa for 48 h at 30 °C, a decrease of about 0.6 log units on the spore counts was observed. Even though the authors did not report the pH of the minced anchovies, it was reported in the literature to be around 6.3–6.4 [37,38], which is close to the pH 6.00 of the present study. In spite of regarding a different *Clostridium* spp., the inactivation level at pH 6.00 in the present study ranged between 0.5 and 0.6 after 48 h under pressure at RT, which aligns with the findings reported by the aforementioned authors.

To place the results obtained in the present work into context, a similar inactivation level was achieved by [29] in beef slurry (pH 6.5) by combining HPP and moderate temperatures (60 °C) for 40 min, with a reduction of almost 2 log units for *C. perfringens* NZRM 898 and 2621 spores, a level of inactivation similar to the one reached after 30 days at 75 MPa/RT in the present study. Even though the abovementioned study regarded HPP, due to the lack of data in the literature focusing the effects of HS on *C. perfringens*, this is the closest comparison possible to make, yet it is understandable that the magnitude of pressures and temperatures used in the study of [29] are far superior to those used in this work. The inactivation observed in HS at pH 6.0 might be hypothesized to be due the pressure stimulus to the germinant receptors (GR’s), leading spores to initiate germination but being not able to conclude the germination process due to the pressure hurdle, resulting in spore death, as previously observed, for example, for *B. subtilis* and *A. acidoterrestris* spores [20,21], with an overall log unit reduction of 4.5 after 30 and 2 days, respectively, under HS at 100 MPa.

At pH 7.50, storage at AP/RT resulted in a significant increase (*p* < 0.05) of the TML (about 2.74 log units), which was accompanied by a small increase (*p* < 0.05) of the HREF (about 0.28 log units) by the fifth day of storage experiments (Figure 3). When it comes to samples under RF conditions, overall, by the 30th day of storage experiments, a decrease of about 0.48 log units was observed. Concerning samples kept under HS at uncontrolled RT, it was observed lower TML reductions when compared to pH 6.00, after 30 days of storage at 75, 150, and 200 MPa, resulted in 1.02, 1.13, and 1.17 log units, respectively, similarly to HREF, with an overall reduction of 1.52, 1.59, and 1.48 log units, respectively.

Considering the promising results obtained with BHI-broth, coconut water (pH value 5.40 and water activity > 0.98) was used as a practical and real case study considering its pertinence in what concerns spores’ development and outgrowth as described in the Introduction section.

Without surprise, inoculated coconut water with *C. perfringens* kept at AP/RT conditions led to a quick endospore germination and outgrowth (Figure 4), resulting in an increase of the TML and HREF of about 1.6 and 1.9 log units, respectively. RF hurdled spores’ development, as the TML by the end of the storage period (30 days) were practically the same (*p* > 0.05) as the initial ones. For example, [17] reported that *Clostridium botulinum* endospore loads remained practically unchanged during 45 days in unprocessed coconut water under RF (4 °C) and 30 days at 10 °C (with growth occurring after 45 days). According to the authors, some natural antimicrobials naturally present in coconut water, such as lauric acid or peptides with antimicrobial activity, could explain the lack of endospore development at both temperatures, as well as the absence of toxin production even under temperature abusive conditions (10 °C) for 45 days. Nevertheless, the spores lost thermal resistance while under RF conditions, considering the reduction of about 1.9 log unit of the HREF after 30 days. This could be due to the ability of the *C. perfringens* spores to start the germination process (which is accompanied by loss of thermal resistance) with the subsequent outgrowth being not able to occur due to the RF hurdle.

When it comes to HS, a gradual spore inactivation was observed, regardless of the pressure level, being more pronounced at higher pressures. Truly, after 30 days of storage at 75, 150, and 200 MPa, an inactivation of about 1.5, 2.1, and 2.9 log units of the TML, respectively, was observed.

The possibility of activating *C. perfringens* spores to germinate using commercial nonthermal HPP at pressures above 300 MPa has been reported in the literature. Summarily, above a certain hydrostatic pressure threshold (above 350–400 MPa), instead of triggering the germinant receptors as for low/moderate pressures (150–200 MPa) and starting the germination process, the DPA channels of the spores are opened, and the spore core hydrates, making them less resistant to further processing [26,39]. Taking this into account, we evaluated the possibility of simulating commercial nonthermal pasteurization conditions by HPP (600 MPa, 3 min, 17 °C) and subsequent HS/RT conditions to control the development of *C. perfringens* spores, as dependent on the pH of the inoculation matrix (BHI-broth) and in coconut water as a case of a real food system, with the results being presented and discussed in the next section.

### 3.2. Impact of HPP and pH on the C. perfringens Spores Behavior under HS Conditions

Right after HPP pasteurization, the initial TML faced a small decrease (Appendix A), which, for all the studied pH values, was no higher than 0.8 log units, possibly due to the presence of some vegetative cells remaining from the harvesting process. In BHI-broth, at pH 4.50, a similar scenario was found for samples pasteurized by HPP compared to the non-pasteurized ones. Generally, there was a small decrease (*p* < 0.05) on the TML for samples kept at AP/RT and under RF. When it comes to samples kept under HS conditions, a decrease on the TML for all the storage conditions, which were more pronounced at 150 MPa with almost 1 log unit reduction, compared to approximately 0.7 log units at 75 and 200 MPa was verified (Figure 5).

Contrarily, the HREF remained practically unchanged under AP/RT and RF, while for samples kept under HS, it a small decrease (*p* < 0.05) was observed compared to the initial values, which were similar (*p* > 0.05) at 75, 150, and 200 MPa after 30 days of storage, with an overall around 0.2–0.3 log units. At pH 6.00, for samples pasteurized by HPP, a similar scenario to that observed for non-pasteurized samples was found. For samples kept under AP/RT, a gradual increase (*p* < 0.05) of both the TML and HREF were observed (0.7 and 0.5 log units, respectively) (Figure 6), while for RF, both TML and HREF were similar (*p* > 0.05) to the initial values kept after 30 days of storage, despite some local variations. A different scenario was observed for samples kept under HS, as a gradual reduction of both the TML and HREF were observed, with an overall TML reduction of about 2 log units, regardless of the storage pressure (75, 150, and 250 MPa) and 1.9 and 2.4 log units at 75 and 150/200 MPa, respectively.

At pH 7.50 (Figure 7), and without surprise, the TML increased 0.5 log units after 15 days of storage at AP/RT, which was accompanied by a decrease on the HREF on the fifth day, followed by an increase. Under RF conditions, there was a general decrease (*p* < 0.05) on the TML of about 0.6 log units (compared to the initial samples), with a similar behavior being observed for the HREF with a 0.22 log unit reduction (*p* < 0.05) compared to the initial loads. The inactivation levels observed for the TML under HS were similar (*p* > 0.05) regardless of the storage pressure after 30 days, with an overall inactivation level ranging between 1.7 and 1.8 log units. A similar trend was found for the HREF, with an overall reduction of about 1.7 log units.

Concerning coconut water when subjected to HPP pasteurization, the TML was reduced by about 0.32 log units (from 5.11 ± 0.11 to 4.79 ± 0.02 Log CFU/mL, as seen in Appendix A). This reduction may have been caused by the inactivation of some vestigial vegetative cells remaining from the harvesting and washing process of the spores and/or due to some spores in the process of germination, as nonthermal HPP is reported to have no effect on spores [29]. In fact, this value was similar to the one obtained to that after the heat shock to determine the HREF at the beginning of the storage experiments (4.82 ± 0.07 Log CFU/mL), which supports this possibility. After 15 days at AP/RT conditions, the spores were able to germinate and outgrow in coconut water right after 15 days of storage, and this increase was also accompanied by an increase (*p* < 0.05) on the HREF (Figure 8), suggesting, as mentioned above, that the vegetative cells were able to sporulate again. Under RF conditions the TML faced an increase (*p* < 0.05) of about 0.33 log units after 15 days. A previous work from [17] in HPP (593 MPa, 3 min, 4 °C) coconut water reported no *C. botulinum* spores’ development under RF (4 °C) conditions for 45 days, although, spores’ development was observed at 10 °C. In another study, [18] studied the effects of HPP (550 MPa, 3 min, 10 °C) and subsequent RF storage in non-toxigenic *C. botulinum* type E and *Clostridium* spp. in coconut water and reported no spore growth under RF (4 °C) and under temperature-abusive conditions (10 °C) for 61 days, which the authors attributed to the lack of specific nutrients for the development of the spores. For HS/RT, a gradual TML decay was observed, with an overall reduction of about 1.3, 1.7, and 1.8 at 75, 150, and 200 MPa, respectively, after 30 days of storage, while minor changes were observed for HREF, with the inactivation levels ranging from 0.18 to 0.55. While it seems to be a long dwell time to achieve this inactivation level, the aim of this preservation methodology (HS) is primarily to hurdle microbial development similarly to that in RF; as such, this inactivation effect is remarkable, especially when it occurs at a low pressure as 75 MPa, and at RT, thus without applying any heat.

The results obtained in this work for *C. perfringens* spores under HS/RT, compared with other results in the literature for other spores, namely *Bacillus* spp., show that *C. perfringens* spores tended to be less responsive towards hydrostatic pressure, as it is the case for suggesting that the nature of the nutrient-receptors of first are less likely to be triggered by hydrostatic pressure compared to the later [40].

### 3.3. Effect of HS on the Inactivation Kinetic Parameters

The experimental data did not fit for most of the cases. The first-order kinetics model, at first visible through the shape of the inactivation curves, and further mathematically modelling confirmed convergence with the Weibull model, as seen in Table 1. There was no consistent pattern regarding the impact of HS, pH and the previous HPP pasteurization (prior to HS) on the inactivation of *C. perfringens* spores, with the exception of pH 6.00, whereas it seemed that the *b* parameter of the Weibull model (regarding the inactivation rate) seemed to be higher for unprocessed samples compared to samples pasteurized by HPP prior to HS, with this being evident at 200 and 250 MPa. A similar trend was observed for coconut water at 200 MPa, wherein unprocessed samples presented a higher *b* value (1.208 ± 0.233) compared to HPP pasteurized samples (0.462 ± 0.132). The graphic representation of both the first-order and Weibull models is displayed in Appendix A, along with the graphical representation of the precited versus experimental values (Appendix A).

The Weibull model was previously reported to generally well describe the inactivation of *C. perfringens* spores by [29], yet this study regarded the combination of HPP with high temperatures (600 MPa, up to 30 min, ≤75 °C) and not HS. This study reported *n*-values (the shape factor) ranging from 0.39 to 0.74, which were overall similar to those reported in the present study, despite a few cases, clearly showing that the inactivation kinetics was nonlinear.

## 4. Conclusions

The present study focused on the feasibility of HS at RT to control the development of *Clostridium perfringens* spores as dependent on pH and on a previous HPP pasteurization step in a model system (BHI-broth) and a real food (coconut water). Reviewing the results, one can conclude that HS was able to inhibit spores’ development, with the clear effect of pH towards the response of the spores while under HS also being evident, especially at pH 4.50 compared to pH 6.00 and 7.50. For the latter pH values, *C. perfringens* spores’ inactivation was observed by HS/RT, while at AP/RT and RF, growth was observed. It was also noted that the inactivation curves were better fitted by the Weibull model when compared to the first-order kinetics model.

Interestingly, globally, HS made it possible not only to inhibit the development of *C. perfringens* spores in a highly perishable food, coconut water but also resulted in their gradual inactivation, regardless of the storage pressure, being more pronounced for unprocessed (raw) than for HPP pasteurized coconut water. The most remarkable result of this work relies on the possibility of controlling the development of *C. perfringens* spores at RT by HS (without temperature control), and in some cases, even inactivating them without the use of any thermal input, as it is generally necessary to inactivate bacterial spores and at pressures low as 75 MPa. This method seems to be promising for extending the shelf-life of foods (by inhibiting spores’ development and even inactivating them along storage), which can be particularly important for those sensitive to thermal processing. Further research could explore the application of HS to other foodborne pathogens and food matrices to expand its utility in food preservation, namely to understand its effects on *Clostridium* spp. spores, and the fundamental inactivation mechanisms.

## Figures and Tables

**Figure 1 foods-13-01832-f001:**
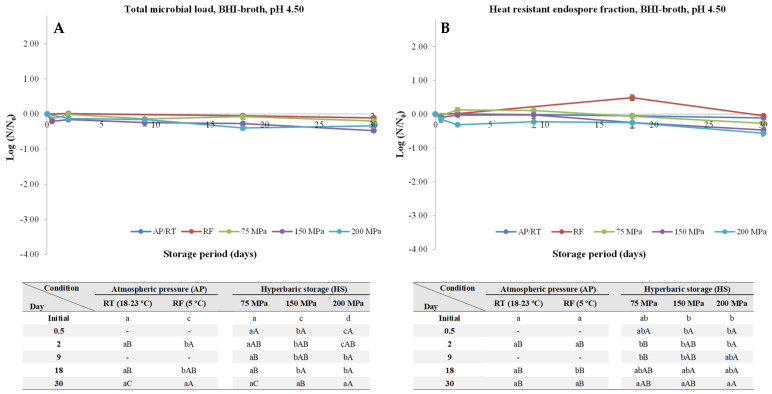
Total microbial load (TML, unheated samples (**A**)) and heat-resistant endospore fraction (HREF, heat-treated samples at 70 °C for 10 min (**B**)) of *Clostridium perfringens* in BHI-broth (pH 4.50) kept at atmospheric pressure (AP) at room temperature (18–23 °C, AP/RT), AP and refrigeration (5 °C, AP/RF), and hyperbaric storage (75, 150, and 200 MPa, HS) at RT, without any pre-activation step. Different lower-case letters (a–d) indicate significant differences (*p* < 0.05) between storage days, while different upper-case letters (A–C) show significant differences between storage conditions (for a given day).

**Figure 2 foods-13-01832-f002:**
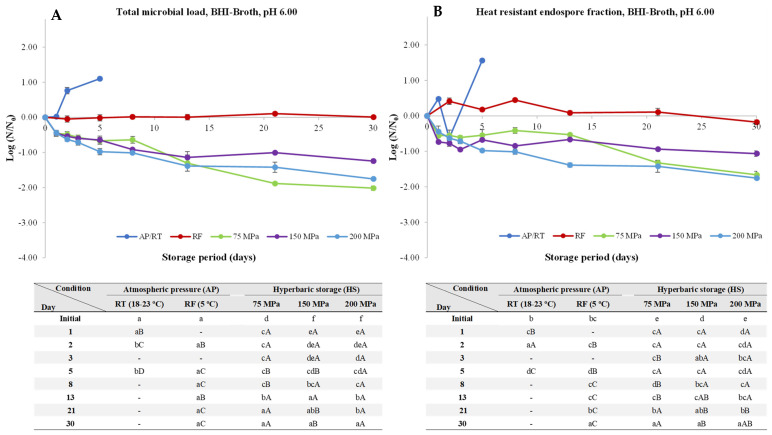
Total microbial load (TML, unheated samples (**A**)) and heat-resistant endospore fraction (HREF, heat-treated samples at 70 °C for 10 min (**B**)) of *Clostridium perfringens* in BHI-broth (pH 6.00) kept at atmospheric pressure (AP) at room temperature (18–23 °C, AP/RT), AP and refrigeration (5 °C, AP/RF), and hyperbaric storage (75, 150, and 200 MPa, HS) at RT, without any pre-activation step. Different lower-case letters (a–f) indicate significant differences (*p* < 0.05) between storage days, while different upper-case letters (A–C) show significant differences between storage conditions (for a given day).

**Figure 3 foods-13-01832-f003:**
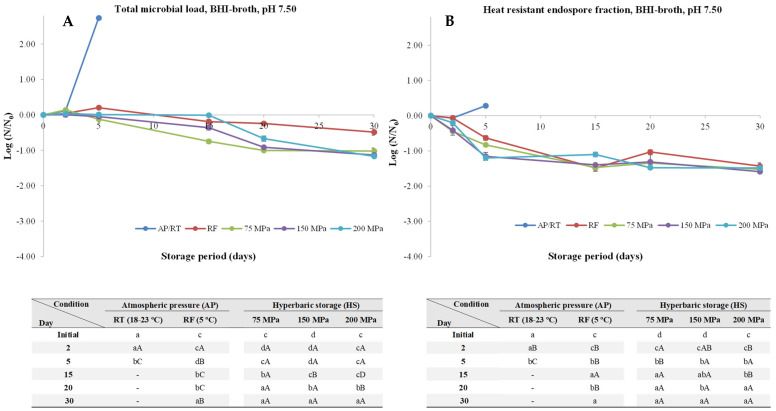
Total microbial load (TML, unheated samples, (**A**)) and heat-resistant endospore fraction (HREF, heat-treated samples at 70 °C for 10 min, (**B**)) of *Clostridium perfringens* in BHI-broth (pH 7.50) kept at atmospheric pressure (AP) at room temperature (18–23 °C, AP/RT), AP and refrigeration (5 °C, AP/RF), and hyperbaric storage (75, 150, and 200 MPa, HS) at RT, without any pre-activation step. Different lower-case letters (a–d) indicate significant differences (*p* < 0.05) between storage days, while different upper-case letters (A–D) show significant differences between storage conditions (for a given day).

**Figure 4 foods-13-01832-f004:**
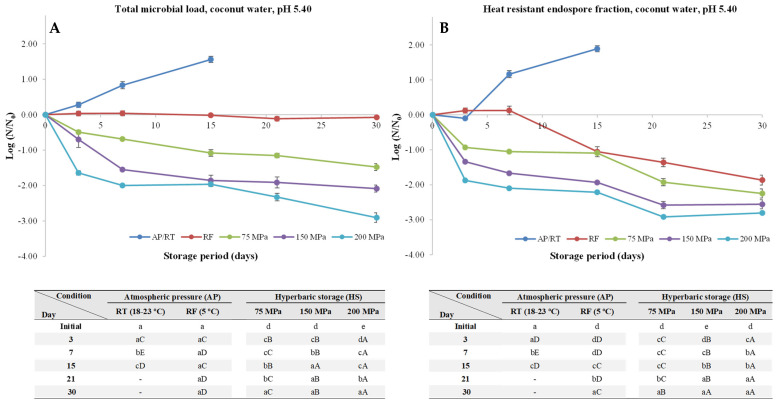
Total microbial load (TML, unheated samples (**A**)) and heat-resistant endospore fraction (HREF, heat-treated samples at 70 °C for 10 min (**B**)) of *Clostridium perfringens* in coconut water (pH 5.40) kept at atmospheric pressure (AP) at room temperature (18–23 °C, AP/RT), AP and refrigeration (5 °C, AP/RF), and hyperbaric storage (75, 150, and 200 MPa, HS) at RT, without any pre-activation step. Different lower-case letters (a–d) indicate significant differences (*p* < 0.05) between storage days, while different upper-case letters (A–E) show significant differences between storage conditions (for a given day).

**Figure 5 foods-13-01832-f005:**
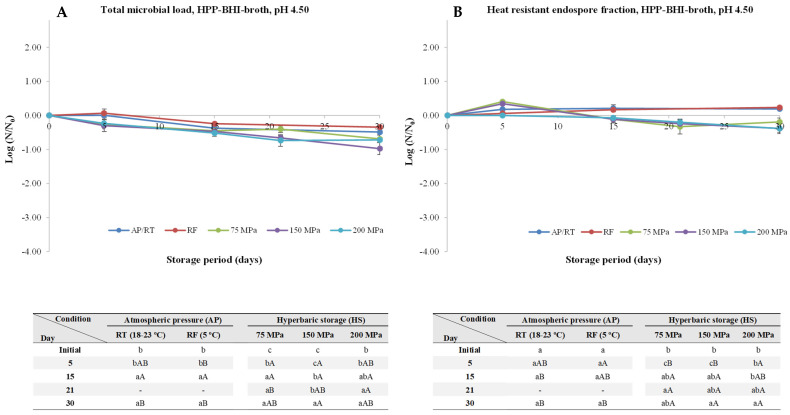
Total microbial load (TML, unheated samples (**A**)) and heat-resistant endospore fraction (HREF, heat-treated samples at 70 °C for 10 min (**B**)) of *Clostridium perfringens* in BHI-broth (pH 4.50) previously pasteurized by high-pressure processing (600 MPa, 3 min, 17 °C) kept at atmospheric pressure (AP) at room temperature (18–23 °C, AP/RT), AP and refrigeration (5 °C, AP/RF), and hyperbaric storage (75, 150, and 200 MPa, HS) at RT, without any pre-activation step. Different lower-case letters (a–c) indicate significant differences (*p* < 0.05) between storage days, while different upper-case letters (A–B) show significant differences between storage conditions (for a given day).

**Figure 6 foods-13-01832-f006:**
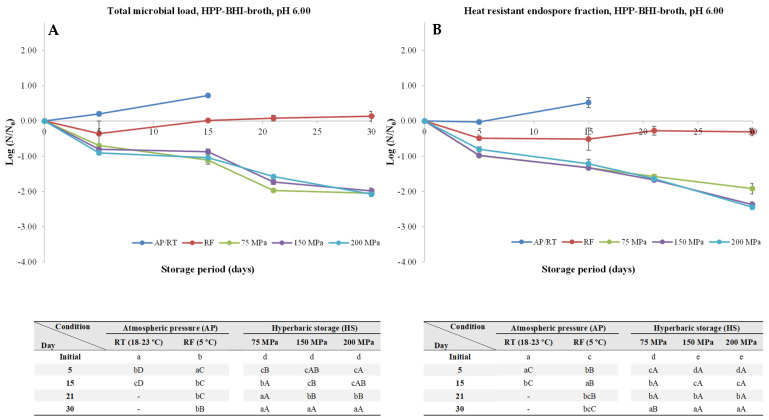
Total microbial load (TML, unheated samples, (**A**)) and heat-resistant endospore fraction (HREF, heat-treated samples at 70 °C for 10 min, (**B**)) of *Clostridium perfringens* in BHI-broth (pH 6.00) previously pasteurized by high-pressure processing (600 MPa, 3 min, 17 °C) kept at atmospheric pressure (AP) at room temperature (18–23 °C, AP/RT), AP and refrigeration (5 °C, AP/RF), and hyperbaric storage (75, 150, and 200 MPa, HS) at RT, without any pre-activation step. Different lower-case letters (a–e) indicate significant differences (*p* < 0.05) between storage days, while different upper-case letters (A–D) show significant differences between storage conditions (for a given day).

**Figure 7 foods-13-01832-f007:**
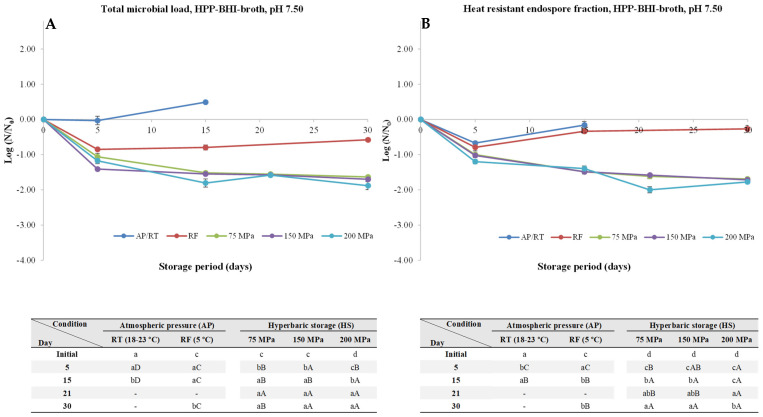
Total microbial load (TML, unheated samples, (**A**)) and heat-resistant endospore fraction (HREF, heat-treated samples at 70 °C for 10 min, (**B**)) of *Clostridium perfringens* in BHI-broth (pH 7.50) previously pasteurized by high-pressure processing (600 MPa, 3 min, 17 °C) kept at atmospheric pressure (AP) at room temperature (18–23 °C, AP/RT), AP and refrigeration (5 °C, AP/RF), and hyperbaric storage (75, 150, and 200 MPa, HS) at RT, without any pre-activation step. Different lower-case letters (a–d) indicate significant differences (*p* < 0.05) between storage days, while different upper-case letters (A–D) show significant differences between storage conditions (for a given day).

**Figure 8 foods-13-01832-f008:**
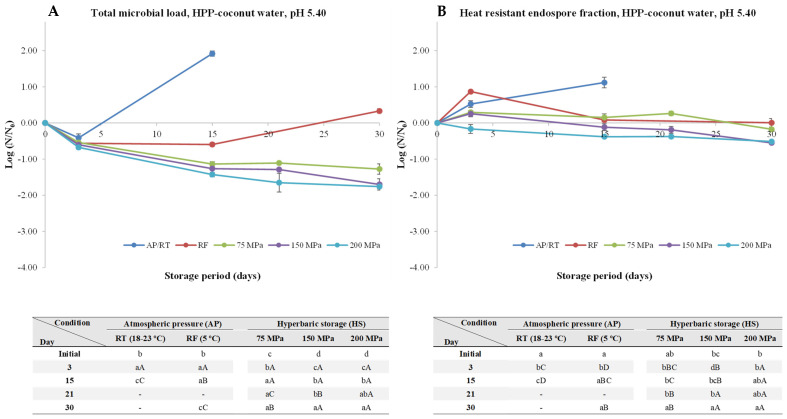
Total microbial load (TML, unheated samples (**A**)) and heat-resistant endospore fraction (HREF, heat-treated samples at 70 °C for 10 min (**B**)) of *Clostridium perfringens* in BHI-broth (pH 6.00) previously pasteurized by high-pressure processing (600 MPa, 3 min, 17 °C) kept at atmospheric pressure (AP) at room temperature (18–23 °C, AP/RT), AP and refrigeration (5 °C, AP/RF), and hyperbaric storage (75, 150, and 200 MPa, HS) at RT, without any pre-activation step. Different lower-case letters (a–d) indicate significant differences (*p* < 0.05) between storage days, while different upper-case letters (A–D) show significant differences between storage conditions (for a given day).

**Table 1 foods-13-01832-t001:** Estimated first-order kinetic parameters (D-, displayed in days) and Weibull model parameters (b and n), displayed as value ± 95% confidence interval, regarding the inactivation of *Clostridium perfringens* spores under hyperbaric storage (HS) conditions (75–200 MPa, 19–23 °C) as dependent of pH level, matrix, and activation method. The * means that it was not possible to fit the model.

Pasteurization Process	Matrix	pH	Pressure (MPa)	First Order Model Parameters				Weibull Model Parameters				
				*D*-Value (Days)	*R* ^2^	Adj-*R*^2^	*MSRE*	b	n	*R* ^2^	Adj-*R*^2^	*MSRE*
No pasteurization	BHI-broth	4.50	75	*				*				
			150									
			200									
		6.00	75	15.53 ± 1.97	0.9135	0.9100	0.1999	0.275 ± 0.073	0.595 ± 0.090	0.9355	0.9329	0.1726
			150	*				0.448 ± 0.065	0.302 ± 0.055	0.9253	0.9223	0.1063
			200					0.490 ± 0.050	0.372 ± 0.037	0.9735	0.9725	0.0874
		7.50	75	*				*				
			150	24.16 ± 3.35	0.9359	0.9318	0.1232	0.014 ± 0.016	1.305 ± 0.349	0.9393	0.9355	0.1199
			200	*				0.001 ± 0.002	2.333 ± 0.728	0.9194	0.9143	0.1392
	Coconut water	5.40	75	*				0.275 ± 0.052	0.489 ± 0.063	0.9821	0.9810	0.0693
			150					0.655 ± 0.199	0.356 ± 0.105	0.9332	0.9290	0.2084
			200					1.208 ± 0.233	0.232 ± 0.069	0.9571	0.9544	0.1977
High pressure processing (600 MPa, 3 min, 17 °C)	BHI-broth	4.50	75	*				*				
			150					0.060 ± 0.057	0.807 ± 0.301	0.9009	0.8932	0.1155
			200					0.108 ± 0.077	0.580 ± 0.231	0.9083	0.9012	0.0954
		6.00	75	14.49 ± 2.56	0.9198	0.9136	0.2363	0.226 ± 0.131	0.661 ± 0.186	0.9475	0.9434	0.1912
			150	*				0.210 ± 0.158	0.656 ± 0.241	0.9108	0.9039	0.2284
			200	16.19 ± 3.04	0.9108	0.9039	0.2242	0.278 ± 0.152	0.573 ± 0.177	0.9397	0.9350	0.1844
		7.50	75	*				0.757 ± 0.113	0.235 ± 0.095	0.9889	0.9880	0.0693
			150					1.193 ± 0.090	0.098 ± 0.026	0.9962	0.9959	0.0420
			200					0.846 ± 0.237	0.235 ± 0.095	0.9622	0.9593	0.1447
	Coconut water	5.40	75	*				0.401 ± 0.108	0.346 ± 0.089	0.9562	0.9528	0.1082
			150					0.369 ± 0.117	0.440 ± 0.104	0.9699	0.9676	0.1128
			200					0.462 ± 0.132	0.405 ± 0.094	0.9721	0.9700	0.1212

* Unable to fit a model.

## Data Availability

The original contributions presented in the study are included in the article/Appendix A, further inquiries can be directed to the corresponding author.

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
