# Peer review of "Impact of pH and High-Pressure Pasteurization on the Germination and Development of *Clostridium perfringens* Spores under Hyperbaric Storage versus Refrigeration"

_foods, 2024, doi:10.3390/foods13121832_

Round 1
Reviewer 1 Report
Comments and Suggestions for Authors
The article describes the influence of physicochemical factors on the germination of Clostridium perfringens, in particular pH, high pressure and temperature. The authors included a comprehensive and clear description of the results of the challenge test using the mentioned parameters. Coconut water and BHI medium were used as matrices. The authors observed the most extraordinary results of their experiments on the above-mentioned matrices for pressures as low as 75 MPa without temperature control. The results of the presented manuscript have application value, especially for food technologists dealing with the problem of inactivation of Clostridium spp. In the text of the manuscript, however, I noticed two formulations that require extension:
Line 38 - strains that are particularly the etiological factor of food poisoning in humans are enterotoxic isolates. Before 2018, it was referred to as C. perfringens type A with the ability to produce enterotoxin (CPE). Currently, these strains are referred to as type F (see Rood et al., 2018).
In the discussion, it would be worth emphasizing the difference in resistance to temperature and physicochemical factors of spores produced by enterotoxigenic strains and those that do not produce CPE (Jaakkola et al., 2021).
Author Response
Referee 1:
The article describes the influence of physicochemical factors on the germination of Clostridium perfringens, in particular pH, high pressure and temperature. The authors included a comprehensive and clear description of the results of the challenge test using the mentioned parameters. Coconut water and BHI medium were used as matrices. The authors observed the most extraordinary results of their experiments on the above-mentioned matrices for pressures as low as 75 MPa without temperature control. The results of the presented manuscript have application value, especially for food technologists dealing with the problem of inactivation of Clostridium spp. In the text of the manuscript, however, I noticed two formulations that require extension:
The authors acknowledge the referees’ valuable inputs, which considerably increased the quality of the manuscript.
Line 38 - strains that are particularly the etiological factor of food poisoning in humans are enterotoxic isolates. Before 2018, it was referred to as C. perfringens type A with the ability to produce enterotoxin (CPE). Currently, these strains are referred to as type F (see Rood et al., 2018).
Changed accordingly.
In the discussion, it would be worth emphasizing the difference in resistance to temperature and physicochemical factors of spores produced by enterotoxigenic strains and those that do not produce CPE (Jaakkola et al., 2021).
Added accordingly. Now it reads, from lines 216-221: “It is also noteworthy that the thermal resistance of C. perfringens spores is quite dependent on their ability to produce enterotoxins, i.e., C. perfringens enterotoxigenic (cpe) strains (those carrying the chromosomal cpe gene) are known to have higher thermal resistance than those lacking the cpe gene. This resistance is primarily due to the presence of a variant of the small, acid-soluble protein (Ssp4), which binds strongly to spore DNA, protecting it from heat damage [31].”.
The authors acknowledge the valuable inputs to improve the discussion of the results.
Reviewer 2 Report
Comments and Suggestions for Authors
The manuscript is very interesting and useful for both scientists and food producers. The authors investigated the effects of pH and high-pressure pasteurisation on the germination and development of Clostridium perfringens spores under hyperbaric and refrigerated storage conditions both under model conditions and using the food product, coconut water (pH 5.40). The abstract, introduction and research methodology has been adequately described. Equations to describe the kinetics of spore inactivation based on two mathematical models are also included: a first-order model and a Weibull model. The results were mainly presented in the form of graphs and tables. Their discussion shows that the use of hyperbaric storage was able to inhibit the development of spores, yoko the effect of pH influence, this effect being most pronounced at pH 4.50. In addition, the use of this method not only inhibited the development of C. perfringens spores in coconut water, but also caused their gradual inactivation. Only the following two points need improvement.
1. Please add two keywords other than those in the title of the manuscript.
2. Latin names of bacterial strains should be written in italics, including C. perfringens. Please correct throughout the manuscript.
Author Response
Referee 2:
The manuscript is very interesting and useful for both scientists and food producers. The authors investigated the effects of pH and high-pressure pasteurisation on the germination and development of Clostridium perfringens spores under hyperbaric and refrigerated storage conditions both under model conditions and using the food product, coconut water (pH 5.40). The abstract, introduction and research methodology has been adequately described. Equations to describe the kinetics of spore inactivation based on two mathematical models are also included: a first-order model and a Weibull model. The results were mainly presented in the form of graphs and tables. Their discussion shows that the use of hyperbaric storage was able to inhibit the development of spores, yoko the effect of pH influence, this effect being most pronounced at pH 4.50. In addition, the use of this method not only inhibited the development of C. perfringens spores in coconut water, but also caused their gradual inactivation. Only the following two points need improvement.
The authors acknowledge the referees’ valuable inputs, which considerably increased the quality of the manuscript.
- Please add two keywords other than those in the title of the manuscript.
Added accordingly.
- Latin names of bacterial strains should be written in italics, including C. perfringens. Please correct throughout the manuscript.
Changed accordingly.
Reviewer 3 Report
Comments and Suggestions for Authors
The article: Impact of pH and high-pressure pasteurization on the germination and development of Clostridium perfringens spores under hyperbaric storage versus refrigeration, is interesting, sent some comments:
1. Review formatting or presentation issues; for example, instead of putting double parentheses, you could put the second parenthesis in a square bracket.
2. In the summary, the authors define abbreviations because they define them again (line 75) and do so in the conclusion (442-445). Review and correct where appropriate.
3. The introduction is very broad. I suggest improving it by including information about the problems of said bacteria and indicating the illness or death it causes in humans and animals. This could emphasize the importance of doing this type of work.
4. Why do they use coconut water? I suggest clearly stating why use it as real food.
5. Improve the conclusion, be careful not to repeat results.
6. The results obtained by the authors are very interesting. However, it is not clear to me what the practical recommendation of the work would be. Would you recommend that foods have hyperbaric storage and the pH be modified?
Author Response
The article: Impact of pH and high-pressure pasteurization on the germination and development of Clostridium perfringens spores under hyperbaric storage versus refrigeration, is interesting, sent some comments:
The authors acknowledge the referees’ valuable inputs, which considerably increased the quality of the manuscript.
- Review formatting or presentation issues; for example, instead of putting double parentheses, you could put the second parenthesis in a square bracket.
Revised accordingly.
- In the summary, the authors define abbreviations because they define them again (line 75) and do so in the conclusion (442-445). Review and correct where appropriate.
Revised accordingly.
- The introduction is very broad. I suggest improving it by including information about the problems of said bacteria and indicating the illness or death it causes in humans and animals. This could emphasize the importance of doing this type of work.
Added accordingly. The requested information was added to the introduction section of the manuscript. Please check the Introduction with the improvements highlighted as red-colour text.
- Why do they use coconut water? I suggest clearly stating why use it as real food.
The referee raised a very pertinent question. Coconut water is a low acidic food product (whose pH varies between 5.2-6.3 in mature state), being so prone to microbial development and presenting no natural hurdles to avoid spores’ development. In this regard, in 2015, HPP coconut water was retrieved from the markets due to safety concerns raised by the US Food and Drug Administration on the possibility of coconut water containing Clostridium botulinum spores and, considering the aforementioned issue regarding pH, refrigeration could only temporarily inhibit the germination and development of C. perfringens spores. Considering the positive results obtained so far regarding the possibility of inhibiting spores’ development by hyperbaric storage, this would make a good example on how this methodology could be used to tackle a real industry case-study. Moreover, the successful application of HS not only to inhibit but also to inactivate C. perfringens spores in coconut water can pave the way for its broader use in preserving other low-acidic food products, or even other heat-sensible value-added products, such as those from the pharmaceutical and biotechnological fields. A clarification on the raised issue was added at the end of the introduction section. The authors acknowledge the valuable insights. Now it reads, in lines 99-105: “The selection of coconut water as a validation case-study relies on the previously mentioned facts that, due to its highly perishability, does not possess natural hurdles against spores’ development, as such, it has a very short shelf-life when it is unprocessed (raw) and refrigeration can only temporarily delay the development of spores (as it was issued by the FDA in 2015). As such, this is an iconic case-study and a good candidate to study the effects of a new preservation methodology (HS/RT) on C. perfringens spores.”.
- Improve the conclusion, be careful not to repeat results.
Changed accordingly. Please check the conclusions’ section with the changed text highlighted as red-colour text.
- The results obtained by the authors are very interesting. However, it is not clear to me what the practical recommendation of the work would be. Would you recommend that foods have hyperbaric storage and the pH be modified?
The results presented in this study aimed evaluate the effectiveness of HS on controlling the development of C. perfringens spores and infer how the spores would behave under HS conditions according to the pH of the inoculation matrix. At pH 4.50, and despite of HS does not inactivate C. perfringens spores (with and without a previous HPP pasteurization step prior to HS) C. perfringens spores are naturally hurdled at this pH level (as the FDA states that below 4.6, C. botulinum spores cannot germinate and develop), as such, HS would only provide an additional hurdle against spore development. For other pHs (6.00 and 7.50) (and 5.40 if we consider coconut water), HS not only inhibited the development of C. perfringens spores but also inactivated them gradually along storage. This means that HS performed equally to better than refrigeration in hurdling spore development (with the plus that HS can inactivate the spores during storage), as such, in a real situation, one would simply store their product under HS at uncontrolled RT without the need to modify the pH.
Reviewer 4 Report
Comments and Suggestions for Authors
The results of the work entitled "Impact of pH and High-Pressure Pasteurization on the Germination and Development of Clostridium perfringens Spores Under Hyperbaric Storage Versus Refrigeration" are good. However, I have some comments that could improve the work before the paper is submitted for further consideration.
Although the authors had a similar study published before in the Foods journal, it focused on another pathogen. What is the difference between the two studies, and where is the novelty in this current work?
There are plagiarized sentences, specifically in the figure legends, which match those in the previously published study in the Foods journal. Please address this issue.
I wonder if this study could be applicable on a large scale or if it is limited to laboratory conditions.
The abstract is too long. Please report only the most important results, as the journal guidelines state that the abstract should not exceed 200 words.
The quality of the figures is very poor. Could the authors reconstruct them in a more professional manner?
The reference "A previous work from Raghubeer et al. (2020) [15]" does not conform to the journal's style guidelines. Please revise all the references accordingly.
Does the journal require the results and discussion to be combined into one section, or should they be separate?
The conclusion currently reads like a summary of the abstract. Please rewrite it to avoid including specific numbers and instead focus on the broader outcomes and implications of the study.
Author Response
The results of the work entitled "Impact of pH and High-Pressure Pasteurization on the Germination and Development of Clostridium perfringens Spores Under Hyperbaric Storage Versus Refrigeration" are good. However, I have some comments that could improve the work before the paper is submitted for further consideration.
The authors acknowledge the referees’ time and efforts to review our manuscript.
Although the authors had a similar study published before in the Foods journal, it focused on another pathogen. What is the difference between the two studies, and where is the novelty in this current work?
The authors acknowledge the issue raised by the referee. The study already published in foods regards ascospores (fungi spores) from Byssochlamys nivea, which is a mould whose spores are quite resistant to thermal processing. The presence of these ascospores in acidic fruit juices, pulps, etc is responsible for juice spoilage and raises some toxicologic issues. If these ascospores are present in foods, and if they are able to germinate and form a mycelium, they will produce patulin, which is a known carcinogenic. From an industrial point of view, this is a concern considering that these ascospores are able to develop under conditions that are quite different from those where bacterial spores develop (namely under acidic conditions).
The current work features the effects of hyperbaric storage on a bacterial spore from the genera Clostridiae (namely the spores from the pathogenic Clostridium perfringens). These spores are also quite heat-resistant and are responsible for food poisoning illnesses when they germinate and develop into a vegetative microorganism. The novelty of this paper regards on the evaluation of hyperbaric storage, as dependent of the pH and of a previous nonthermal pasteurization process by HPP prior to hyperbaric storage, to test the feasibility of hyperbaric storage to hurdle spore development. To the authors best knowledge, this is the first time that the effects of hyperbaric storage on a spore from the genera Clostridiae are evaluated.
There are plagiarized sentences, specifically in the figure legends, which match those in the previously published study in the Foods journal. Please address this issue.
Revised accordingly. The figure legends were done on a basis of providing all the information required to fully understand the information displayed in each one without the need to check the main text (manuscript). This way, the legends are rather big and indeed contain similarities to those observed in a previous paper that we are the authors (reference below), for the sake of easiness writing. Indeed, the experimental conditions were similar, regarding the storage conditions, this resulting in similarities in the figures’ legends. Yet, the microorganisms are different, and present different pressure-resistances and each article address different issues with each microorganism.
(Pinto, C. A., Galante, D., Espinoza-Suarez, E., Gaspar, V. M., Mano, J. F., Barba, F. J., & Saraiva, J. A. (2023). Development control and inactivation of Byssochlamys nivea ascospores by hyperbaric storage at room temperature. Foods, 12(5), 978., https://doi.org/10.3390/foods12050978)
I wonder if this study could be applicable on a large scale or if it is limited to laboratory conditions.
Currently, there are no commercial applications of hyperbaric storage at an industrial level, at least as the authors are aware. Regarding the application of hyperbaric storage in a larger scale, it could be possible to apply this methodology at an industrial level, depending on the capacity of pressure vessels, or the pipelines (for example, if one would like to store a pumpable liquid in a pipeline, it would disregard the use of a pressure vessel, instead, the product would be kept under pressure in a network of pipelines, using the liquid product itself as a pressurization fluid. Moreover, for the cases where either the liquids are not pumpable, or the process requires storing solid food products, conventional HPP equipment could be used (as no specific pressure equipment is available for hyperbaric storage applications, which implies that either HPP equipment are used (with large and thick vessels designed to be operated up to 600 MPa) or specifically designed (custom made) equipment is used. The results from this study though provide the first insights on the effectiveness of HS to control the germination and development of C. perfringens spores in a model system, which was afterwards tested in a highly perishable food product. As such, this application can be used beyond laboratorial conditions.
The abstract is too long. Please report only the most important results, as the journal guidelines state that the abstract should not exceed 200 words.
The abstract was reduced to 239 words. As the amount of data available in the manuscript is quite extensive, it was indeed difficult to shorten even more the abstract without loosing accuracy and context. As such, the authors did their best efforts to reduce the abstract as much as possible.
The quality of the figures is very poor. Could the authors reconstruct them in a more professional manner?
The figures where reintroduced in the text and amplified to improve the quality. Thank you for bringing this issue to our attention.
The reference "A previous work from Raghubeer et al. (2020) [15]" does not conform to the journal's style guidelines. Please revise all the references accordingly.
Changed accordingly.
Does the journal require the results and discussion to be combined into one section, or should they be separate?
Thank you for raising this issue. The journal now allows to combine the sections of Results and Discussion in the same section. As this combination makes the presentation of results and further discussion, allows to add fluidity to the text and avoid repeating ideas and references, as such, the authors decided to combine both sections.
The conclusion currently reads like a summary of the abstract. Please rewrite it to avoid including specific numbers and instead focus on the broader outcomes and implications of the study.
Changed accordingly. The conclusion section was revised.
Round 2
Reviewer 4 Report
Comments and Suggestions for Authors
The authors addressed all my concerns effectively. I believe the manuscript can be accepted for publication in Foods Journal.